# Anomaly Detection and Concept Drift Adaptation for Dynamic Systems: A General Method with Practical Implementation Using an Industrial Collaborative Robot

**DOI:** 10.3390/s23063260

**Published:** 2023-03-20

**Authors:** Renat Kermenov, Giacomo Nabissi, Sauro Longhi, Andrea Bonci

**Affiliations:** Department of Ingegneria dell’Informazione (DII), Università Politecnica delle Marche, 60131 Ancona, Italy

**Keywords:** anomaly detection, collaborative robotics, fault detection, concept drift adaptation, machine learning

## Abstract

Industrial collaborative robots (cobots) are known for their ability to operate in dynamic environments to perform many different tasks (since they can be easily reprogrammed). Due to their features, they are largely used in flexible manufacturing processes. Since fault diagnosis methods are generally applied to systems where the working conditions are bounded, problems arise when defining condition monitoring architecture, in terms of setting absolute criteria for fault analysis and interpreting the meanings of detected values since working conditions may vary. The same cobot can be easily programmed to accomplish more than three or four tasks in a single working day. The extreme versatility of their use complicates the definition of strategies for detecting abnormal behavior. This is because any variation in working conditions can result in a different distribution of the acquired data stream. This phenomenon can be viewed as concept drift (CD). CD is defined as the change in data distribution that occurs in dynamically changing and nonstationary systems. Therefore, in this work, we propose an unsupervised anomaly detection (UAD) method that is capable of operating under CD. This solution aims to identify data changes coming from different working conditions (the concept drift) or a system degradation (failure) and, at the same time, can distinguish between the two cases. Additionally, once a concept drift is detected, the model can be adapted to the new conditions, thereby avoiding misinterpretation of the data. This paper concludes with a proof of concept (POC) that tests the proposed method on an industrial collaborative robot.

## 1. Introduction

Industrial collaborative robots (cobots) are designed to work near humans or cooperate with them. They can be easily reprogrammed, e.g., via drag-and-drop programming; they have become widely adopted in smart factories for many tasks, such as assembly, manipulation, polishing, pick and place, etc. [1]. Due to the widespread and consequent economic repercussions in the industrial ecosystem, it is crucial to develop an effective fault detection algorithm that can provide early warnings of imminent faults. In this way, it can avoid further damage and enable better maintenance planning and logistics. One possible approach to diagnose malfunctioning in such systems is through Anomaly Detection (AD). AD refers to the problem of finding patterns in data that do not conform to expected behaviors [2]. The applications are split into fields such as intrusion and fraud detection, medical monitoring, machine vision, etc. In the robotics context, data anomalies may be symptoms of faults, thus AD has been widely studied with the meaning of fault detection [3,4]. The given AD algorithms are generally applied assuming that working conditions are bounded, but cobots are generally used in flexible manufacturing where they perform dynamic tasks. This means that the same cobot can be used in a single working day for different tasks, executing two, three, or more programs. For this reason, the changes in data patterns may refer to actual anomalies as well as behavior changes in the executed program (e.g., task changes or working environment changes). In this article, we distinguish two kinds of changes in data patterns:Anomalies: Equipment faults or defects where the most common are basically of a mechanical nature.Concept drift [5] (CD): Changes related to the program executed. In dynamically changing and nonstationary environments, the data distribution can change over time for many different reasons. In this manuscript, the CD phenomenon has been associated with the definition of a behavioral change in the robot-executed program (different speeds, trajectories, payloads, etc.).

Thus, algorithms for AD in such cases have to distinguish between a failure vs. a CD; in this way, we are able to develop an unsupervised anomaly detection (UAD) system that can work under a CD. When the change of a data pattern is caused by the task change of the robot (or external environment), it is identified as CD. When no CD is detected, the change in the data pattern is identified as an anomaly. Such a system has the following characteristics:1.It can detect an anomaly with a model-free approach;2.It can detect a CD with an unsupervised data-driven approach;3.Once one of the two phenomena is detected, it reacts appropriately. If an anomaly is detected, the system will send a warning, otherwise, in the case of CD detection, the condition monitoring system will adapt to the new working conditions (concept drift adaptation [6]).
To achieve these purposes, the system uses two models: an anomaly detector and a CD detector. The scheme of how the two models work is shown in Figure 1.

**Stage** **1:**training stage. During this stage, both models are trained through the data stream.**Stage** **2:**CD detection stage. This stage is based on a machine learning model called the CD detector. When a CD is detected, both the anomaly detector and the CD detector return to the training stage (CD adaptation). If no CD is detected, the anomaly detection stage begins. Further details will be given in Section 3.1.**Stage** **3:**Anomaly Detection stage. During this stage, the AD model compares the nominal data functioning with the real-time data stream, calculating residuals to monitor cobot conditions. More details will be given in Section 3.2.

The remainder of the paper is organized as follows: Section 2 introduces a literature overview. In Section 3, we present the methodology used to develop the idea exposed during the introduction. Section 4 is devoted to presenting the experimental setup and results. Finally, the paper’s conclusions along with future works are summarized in Section 5.

## 2. Related Work and Background

In the context of robotics systems, we can distinguish three approaches to fault detection [7]: data-driven, model-based, and knowledge-based. The model-based approach [8,9] relies on an accurate mathematical model that can predict the correct behavior of the system, which is compared with the observations. Knowledge-based approaches usually relate recognized behaviors with predefined known faults. For instance, in [10], the authors recognize the faulty frequencies in the current spectrum of industrial packaging machinery. An overview related to this approach can be found in [11]. The major issue of both model-based and knowledge-based approaches is the need for prior knowledge of a model (monitored system in the first case, fault in the second). On the other hand, the data-driven approach does not need any models. Instead, it requires a large amount of labeled data for supervised algorithms or relevant computational power for real-time learning in unsupervised algorithms. One possible solution to this problem was proposed by Park et al. [12] and was reviewed in our previous work [13]. In particular, Park et al. constructed a data model that can hierarchically analyze the relationships between sensing values and cobot operation information, analyze data correlations between the sensing data and operation data to track programmable motions with anomalies, and define the detection criteria of a programmable motion-fault by statistically analyzing the sensing values with the same program and motion. Similarly, we propose a data labeling and indexing method to trace cobot trajectories every time a new program is defined and analyze motion discrepancies in healthy and unhealthy conditions. However, both these approaches could struggle when the motion properties of the robot vary, although the program remains the same (i.e., non-deterministic trajectories, unexpected collision with operators, or varying payloads).

Among the works that have used unsupervised data-driven approaches, Cheng et al. [14] used the Gaussian mixture model to cluster normal and abnormal conditions from the current signal of a robot manipulator. However, their technique can detect faults that are already known. In [15], the authors used a Sliding Window–Convolutional Variational AutoEncoder (SW–CVAE) for the online UAV of an industrial manipulator. The authors could monitor the state of the machine without any prior knowledge about the anomaly. Nevertheless, they did not take into account that the manipulator could change its functioning (CD). Many others have obtained similar results with different approaches. For example, Khalastchi et al. [3,16] monitored the conditions of autonomous robots using the Mahalanobis distance [17] and Azzalini et al. [18] proposed a minimally supervised (where very few nominal observations are required) anomaly detection approach for autonomous robots via a Variational Autoencoder (VAE), which was incrementally trained online. In [19], the authors compared different AD methods on a collaborative robot. To the best of our knowledge, no one faced the CD problem in the unsupervised anomaly detection of robot manipulators; this is the main contribution of this work.

### 2.1. Probabilistic Encoder

The CD detector was developed using a Neural Network (NN), which is part of a VAE [20], called a Probabilistic Encoder (PE). In the literature, we can find many works that use generative models for fault detection or anomaly detection purposes [15,18,21,22]. Typically those works use the value of the reconstruction error of the model to evaluate the health of the system. We trained our model to a known distribution. The output of the model should be a value near that distribution, and the distance between such a distribution and the output is used to evaluate the presence of a CD. The PE maps the input vector $x\in R^{d}$ into the output vector $z\in R^{l}$, as shown in Figure 2. The output vector z is a sample of a posterior distribution $p_{\theta }\left(z|x\right)$ where θ are the weights of the Encoder. We assume that the distribution $p_{\theta }\left(z|x\right)$ is Gaussian and we want it to be as close as possible to a known distribution (in our case, a standard Gaussian distribution N(0,I)). Kullback–Leibler [23] (K–L) divergence is a metric of how much one probability distribution is different from the second; we use it as the loss function:$$L\left(\theta \right)=D_{KL}(p_{\theta }\left(z|x\right)||N\left(0,I\right)).$$

The latter should be minimized by finding the optimal weights θ using the backpropagation algorithm.

### 2.2. Concept Drift

In dynamically changing and nonstationary environments, the data distribution can change over time, yielding the phenomenon of concept drift. This is a crucial concept in robotics contexts. Cobots are used in highly dynamic tasks and different industrial scenarios. Developing a condition monitoring strategy to detect anomalies in cobot data without considering the concept drift can result in difficulties in consistently identifying faults in all possible system motions. This is due to motion discrepancies in the various tasks accomplished (different positions, speeds, loads, etc.). Lane et al. [24] faced a similar problem in the computer security domain. The authors had to detect anomalous or suspicious behaviors in the user’s profile. At the same time, they considered that the behavior characteristic of the nominal user evolves, i.e., the CD. The CD has been faced by many different authors in different contexts [25]. This paper applies CD to the problem of anomaly detection for cobots that frequently change tasks or environments.

## 3. Methodology

The objective of this article is based on the analysis of the cobot’s data stream. Since it is possible to access data related to the position, speed, torque, current, and many other measurements from the cobot’s controller, we use multivariate time series data for the CD detector.

Differently, to address and detect anomalies, it only used torque information because the authors demonstrated in previous work [8,26] that the torque signal contains strong faulty features related to the motor and the kinematic chain attached to the shaft.

In the following paragraphs, we describe the CD detector and fault detector in more detail. Moreover, we will put it all together by proposing a robot operating system (ROS)-based framework.

### 3.1. Concept Drift Detector

To capture the nonlinearity and the temporal dependency of the multivariate sensor variables, we apply the sliding-window technique and a convolutional neural network (CNN) [27] to the encoder (Figure 2). We call this model the sliding window–probabilistic encoder (SW–PE). At each time step *t*, we run an overlapping sliding window with a length of ω and each sample in this sliding window is an *n*-dimensional array. Therefore, the input of the model is an (ω,n)-dimensional matrix. As we mentioned before, our SW–PE is trained on a fixed distribution N(0,I). During the CD detection stage, the model is fed by the data stream. Each sliding window corresponds to an output and each output corresponds to a score, as shown in Figure 3. The score is calculated using the Mahalanobis distance between the distribution N(0,I) and the output vector of our model z:$$d_{M}\left(N\left(0,I\right),z\right)=\sqrt{z^{T}Iz}.$$

During the training stage, we need to evaluate a threshold score. When an output exceeds this threshold value, a CD is detected. Additionally, a perfect CD detector is insensible to faults. Both nominal functioning and nominal functioning with a fault must satisfy the following condition:$$y\leq CD_{th},$$
where y is the set of observed values (CD score) and $CD_{th}$ is the threshold. In an ideal case, we could define $CD_{th}$ as the maximum value of the observations:$$CD_{th,0}=\sup y_{obs}.$$

As we are dealing with a limited amount of data, uncertainties are unavoidable. Therefore, we add a bias to the ideal threshold ${CD}_{th,0}$:$$CD_{th}=CD_{th,0}+b,$$
where $b=rCD_{th}$, and *r* is a parameter between 0 and 1, chosen in an empirical way. Finally combining the last two equations, we obtain:(1)$$CD_{th}=CD_{th,0}\left(1+r\right).$$

### 3.2. Anomaly Detector

During the training stage (Figure 1), the detector acquires a sequence of normal functioning data. In the fault detection stage, these data are compared with the real-time data stream using residuals:$$r_{i}\left(t\right)=T_{i,ref}\left(t\right)-T_{i,obs}\left(t-\hat{D}\right),$$
where $T_{i,ref}$ is the nominal torque profile, $T_{i,obs}$ is the real-time torque acquisition, *i* is an integer number associated with the respective joint, and $\hat{D}$ is the estimated delay between the signals. Ideally, $T_{i,ref}\left(t\right)=T_{i,obs}\left(t\right)$, and $r_{i}\left(t\right)=0$ if the fault does not appear. However, the real data contain different disturbances; thus,
(2)$$\begin{array}{cc} T_{i,ref}\left(t\right) & =T\left(t\right)+n\left(t\right)\;\;\;\;0\leq t\leq t_{n}\\ T_{i,obs}\left(t\right) & =T\left(t-D\right)+n\left(t\right)\;\;\;\;0\leq t\leq t_{obs} \end{array}$$
where T(t) is the ideal profile, $t_{obs}$ is the time window of the observations, $t_{n}$ is the time window of nominal data acquisitions, and n(t) is the sum of all disturbances, which we assume to be white Gaussian distribution. Moreover, to estimate the time delay, we maximize the cross-correlation of the two signals:$$\hat{D}=\underset{\tau }{\mathrm{argmax}}R_{T_{i,ref},T_{i,obs}}\left(\tau \right),$$
where
(3)$$R_{T_{i,ref},T_{i,obs}}\left(\tau \right)=\sum_{t=\tau }^{T_{obs}+\tau }T_{i,ref}\left(t\right)T_{i,obs}\left(t-\tau \right).$$

During the monitoring, we compute the mean value of $r_{i}\left(t\right)$ over the observation period and compare that value with a threshold $F_{th}$. If that happens,
$$|\bar{r}|>F_{th}$$
a fault is detected. The value of the threshold is established by observations as we saw in the CD detector.

### 3.3. ROS-Based Condition Monitoring Architecture

As a real industrial cobot was used as a case study in this paper, the authors developed a ROS package to establish a connection with the cobot controller, program the cobot, execute trajectories, and perform data acquisition. The entire ROS-based condition monitoring architecture used in this work to acquire and preprocess data during the normal functioning of cobots was already presented in previous work [13], where the benefits and limitations in the application of automatic fault detection algorithms in collaborative robots were reviewed. This paper extends the work by applying an unsupervised machine learning model (i.e., the PE) to automatically distinguish each time the cobot changes its task. For the testing part, we developed two different programs that simulated the standard working functions of a cobot. These experiments were conducted to identify anomalies in cobot data or variations in the executed program (considered as CDs).

## 4. Experimental Setup and Results

In order to demonstrate the effectiveness of the proposed method, we provide experimental results using an industrial cobot. Both the CD detector and the anomaly detector were tested on three different data acquisitions: the first one was a nominal program in which the models were trained; the second one was a new program executed with different trajectories, speeds, and payloads; the third one was a nominal program with fault simulation. The next subsection discusses how the fault was simulated on the nominal data stream. The Omron TM5-900 cobot was used in this work to obtain the data acquisitions (Figure 4): for each joint, we acquired positions and torques at 40 Hz. Its specifications are listed in Table 1. The CD detector was tested with all three datasets. We will go into more detail about this in Section 4.3. We will discuss the evaluation of the anomaly detector in Section 4.4.

The motion commands can be sent to the cobot using a programmable logic controller (PLC) or ROS as discussed in our previous work [13]. Providing detailed information on how to define motion commands using ROS and the trajectory planners library is beyond the scope of this paper; therefore, detailed information can be found via the GitHub read me file regarding the usage of our repository [28].

### 4.1. Fault Simulation

In this work, in order to test the proposed algorithm, we simulated faulty behavior manipulating real cobot data acquisition. This subsection proposes a brief overview of mechanical defect modeling for systems driven by electric motors (such as cobots), which is useful for understanding how the faulty condition was introduced in the data. A cobot, from a constructional point of view, can be viewed as a system composed of electric motors and speed reducers (which compose a joint) connected between them, recreating a complex kinematics chain. Therefore, when we address problems related to a cobot joint malfunction, we can exploit considerations derived from the electric motor anomaly detection background. In particular, a defect in the motor or the transmission system (e.g., a speed reducer) attached to the motor shaft causes load torque oscillations [29]. This anomalous behavior can be modeled as the sum of a constant component ($T_{const}$, which represents the load of the motor) and a time-varying term, which depends on the characteristic frequency $f_{c}$ (which represents an order of the rotational frequency of the rotor $f_{r}$). The additional time-varying term, which represents a torque oscillation, arises when the system is working under defects; it is generally modeled through a Fourier series by neglecting higher-order terms [29]:(4)$$T_{l}\left(t\right)=T_{const}+T_{c}\cos\left(\omega_{c}t\right)$$
with $T_{c}$ being the amplitude of the load torque oscillations and $\omega_{c}=2\pi f_{c}$ being the term containing the fault characteristic frequency $f_{c}$. Respectively, the mechanical equation of an electric motor under mechanical defects can be written as:(5)$$\frac{d\omega_{r}}{dt}=\frac{1}{J}\left[T_{e}\left(t\right)-B\omega_{r}-T_{const}-T_{c}\cos\left(\omega_{c}t\right)\right]$$
with *J* being the rotor inertia, $T_{e}$ is the electromagnetic torque, *B* is the viscous friction, and $\omega_{r}$ is the rotor speed. Based on these considerations, some papers propose facing joint fault detection in robotics by using torque information. Robotics manipulators (i.e., cobots) may fall into fault statuses if involved in long heavy manipulations, leading to unexpected production stops because the cobot loses its repeatability and accuracy; consequently, the industrial tasks cannot be accomplished. Therefore, monitoring cobot joint torques in real time may be a valid solution to avoid unpleasant working stop situations. In this regard, Izagirre et al. [30] proposed performing a joint root cause analysis through the study of the torque signal. They experienced a homogeneous increment in the torque when a malfunction in the mechanism of the robot joint arose. Regarding mechanical defects, the torque value increased in the whole trajectory, not only in specific movements or positions. This is because mechanical deterioration creates friction in the joint mechanism affecting the entire movement of the joint. Therefore, when the robotic system works in healthy conditions, we have ($T_{l}\left(t\right)=T_{const}$), i.e., the motor of the joint produces a certain electromagnetic torque during normal functioning to execute the motion request. Differently, when a malfunction arises that produces torque oscillations ($T_{l}\left(t\right)=T_{const}+T_{c}\cos\left(\omega_{c}t\right)$), the motor compensates the mechanical unbalance through torque ($T_{e}$) to accomplish the same task. These theoretical considerations were also experienced in [30,31], where the torque signal was used to perform a cause analysis of faulty joints. In this regard, in this work, to simulate a mechanical malfunction in a cobot joint, in real torque data acquisitions, we added a homogeneous random distributed increment of 10% of the joint torque value. This randomly distributed increment was applied to the torque data of the cobot’s first joint. We simulated a mechanical defect in the first joint since it is one of the most involved in all of the trajectories executed by the cobot. Moreover, the idea was to prove the effectiveness of the proposed method under a highly dynamic context (in data with much variability). Therefore, the objective of this paper is to define an algorithm that can distinguish anomalies related to malfunctions and the CD. In the first case, it is important to have an algorithm that can isolate and detect the abnormal behavior by monitoring joint torques, while in the latter, the algorithm should be able to understand that the system’s working conditions are changed (and the NN has to perform a new training phase).

### 4.2. Encoder Structure and Training of the CD Detector

At each time step *t*, a sliding window of length ω=151 and an overlap of L=111 are applied. Each sample is a vector of dimension n=12 composed of the positions and torques of each joint (6 positions and 6 torques). Therefore, the input of the encoder is a 12×151 dimension matrix. The encoder consists of two 2D convolutional layers followed by a fully-connected layer. The first convolutional layer uses 4 filters and the second uses 16 filters with a length of 3 and stride of 1. After every convolutional layer, we performed batch normalization and applied a rectified linear unit (ReLU) as nonlinearity. The structure is summarized in Figure 5.

For training and testing, we acquired about 3.7 min of data streams for the current program (8998 samples) and 1.2 min of data streams for the new program (2998 samples). Then, we standardized the data to zero mean and unit variances. To train the model, we set the batch size to 48 and the learning rate to $10^{-3}$ in the Adam optimizer.

### 4.3. CD Detector Evaluation

The SW–PE was trained over three minutes of normal functioning. After the training stage, the critical part was the choice of the threshold. Considering Equation (Equation 1) and setting parameter *r* to 7%, we obtained the value of threshold $CD_{th}∼6.7$. Figure 6 represents the scores of normal functioning and Figure 7 represents the scores of normal functioning with faults on Joint 1. As we can see, all of the scores are under the red line (which represents the threshold) and no false CDs are detected. In Figure 8, we have the score of the new program from when the CD occurred. This time, the CD is detected in 12 out of 67 timesteps. This is reasonable because every temporal window is ∼3.8 s of the motion, and although the overall program has changed, it still contains movements that have the same features as in the previous program. For our purpose, it is enough that the CD is detected at least once.

### 4.4. Fault Detector Evaluation

To evaluate the fault detector, we considered the torque signal of joint 1. First, we settle the parameters of the time windowing of Equation (Equation 2):$$t_{n}=25\;s$$
$$t_{obs}=5\;s.$$
The acquisitions of nominal and observation data are shown in Figure 9.

Using Equation (Equation 3), we find the estimated time delay $\hat{D}$ and then we associate the observation with the corresponding part of the nominal graph, as shown in Figure 10.

Finally, we calculate the residuals and evaluate the threshold. In the absence of fault, the residuals will have the form of white noise with zero means. When the fault occurs, we notice a bias that we could detect by using a threshold, as seen in Section 4.4. In Figure 11, we present the plot of residuals under health conditions and when the fault occurs. The red line represents the mean and the blue band represents the healthy zone. The latter is delimited by the threshold $F_{th}$. As shown in the figure, when the fault occurs, the red line moves out from the healthy zone. Therefore, we can clearly distinguish between when the cobot is working under nominal conditions (the nominal program) and when the cobot is executing a different program (CD) (and a new training phase is required). Moreover, we can identify anomalies in data acquisition that can be related to a mechanical defect in the system. In particular, in these experiments, a mechanical defect in the first joint was simulated by recreating its effect on the torque signal.

## 5. Conclusions

In this work, we examined the main problem of UAD in industrial cobots, which is their extreme versatility that results in unpredictable changes in the data stream acquired from the controller. We named this phenomenon ’concept drift’ and proposed an anomaly detection algorithm that can work under a CD and dynamically adapt every time the working conditions change. The approach can clearly distinguish between anomalies and CD in a robot-executed program. In our previous work, we built the entire architecture to acquire data streams using ROS and by executing cobot motions. This paper focused on analyzing data coming from the real system through UAD. In particular, this paper extends previous results by applying an unsupervised machine learning model (i.e., the PE) to automatically distinguish between the cobot’s normal functioning, when a different program is executed, and when there are data anomalies related to possible mechanical defects. The experimental results obtained are encouraging and will motivate further research in this direction. Further investigations are needed to obtain a robust and reliable model that tests the proposed solution in different industrial applications. We used two different models—one for CD detection and the other for anomaly detection. A future step will be to build a unique model with the same capabilities as the two proposed in this paper, in order to address the computational power limitation, which is one of the main drawbacks of our technique.

## Figures and Tables

**Figure 1 sensors-23-03260-f001:**
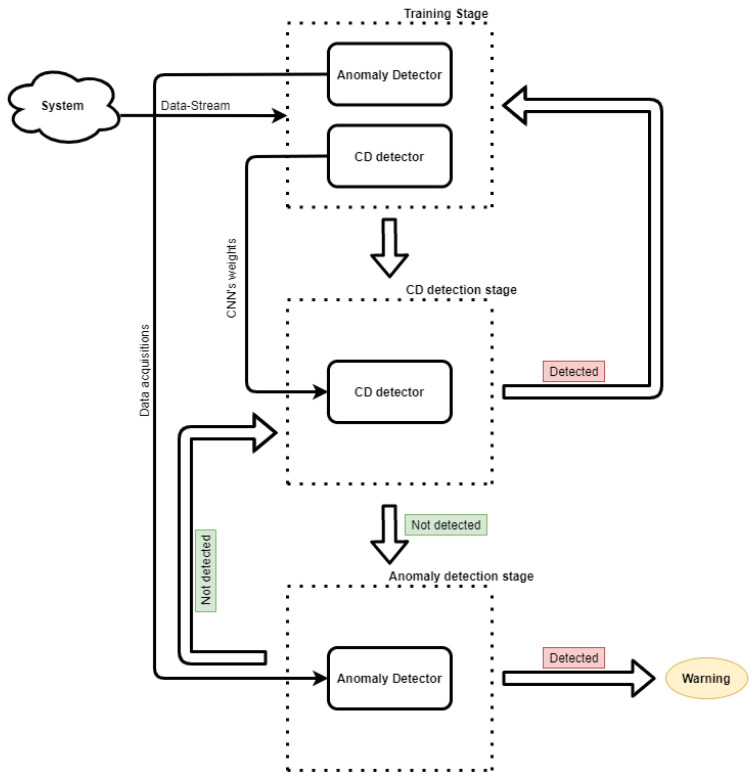
Conceptual scheme of the adaptive anomaly detector. The double arrow indicates the stage transition while the single arrow indicates the data flow. In particular, from the training stage to the CD detection stage, the data that flow are the weights of the CD detector model; from the training stage to the anomaly detection stage, the data that flow are the raw data selected from the data stream.

**Figure 2 sensors-23-03260-f002:**
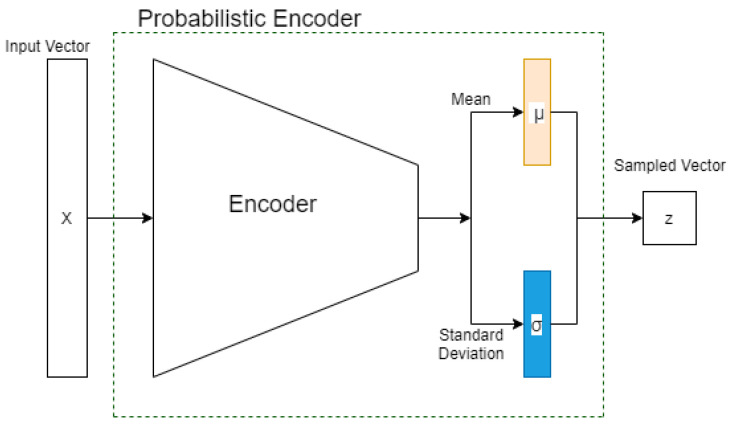
General structure of a probabilistic encoder.

**Figure 3 sensors-23-03260-f003:**
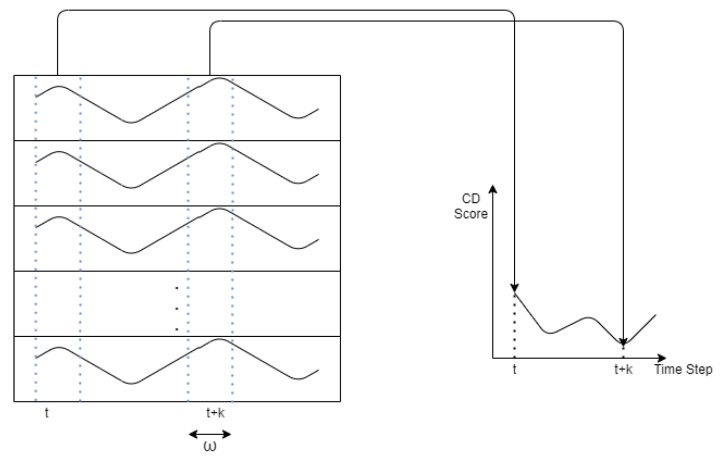
Concept drift detector scheme: it takes a temporal sliding window as input and has a score as the output.

**Figure 4 sensors-23-03260-f004:**
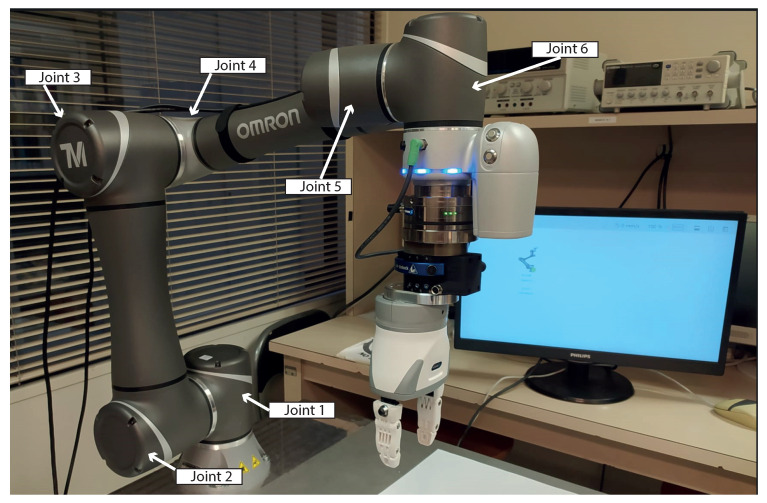
Image of the Omron TM5-900 collaborative robot used for the case study.

**Figure 5 sensors-23-03260-f005:**
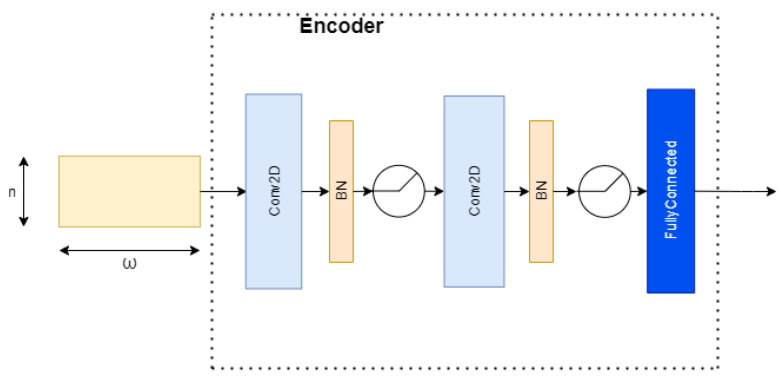
Encoder CNN structure.

**Figure 6 sensors-23-03260-f006:**
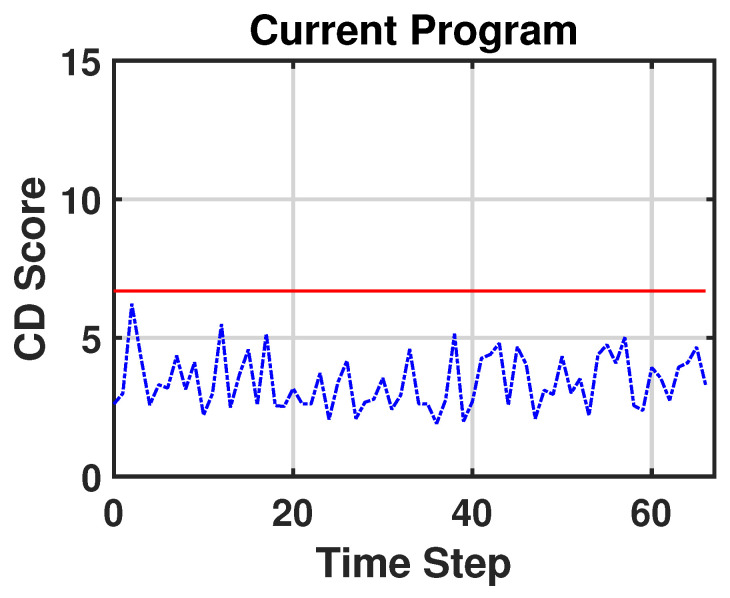
CD scores of the current program without an anomaly.

**Figure 7 sensors-23-03260-f007:**
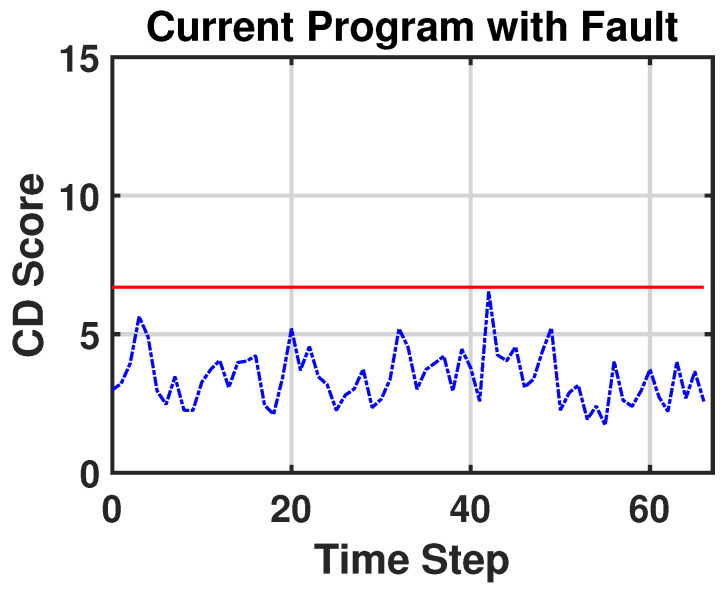
CD score of the current program when an anomaly occurs.

**Figure 8 sensors-23-03260-f008:**
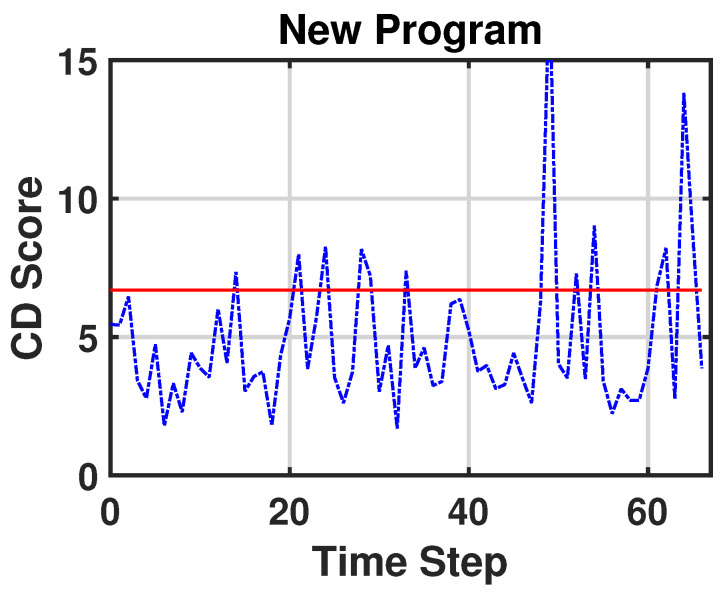
CD score of the new program.

**Figure 9 sensors-23-03260-f009:**
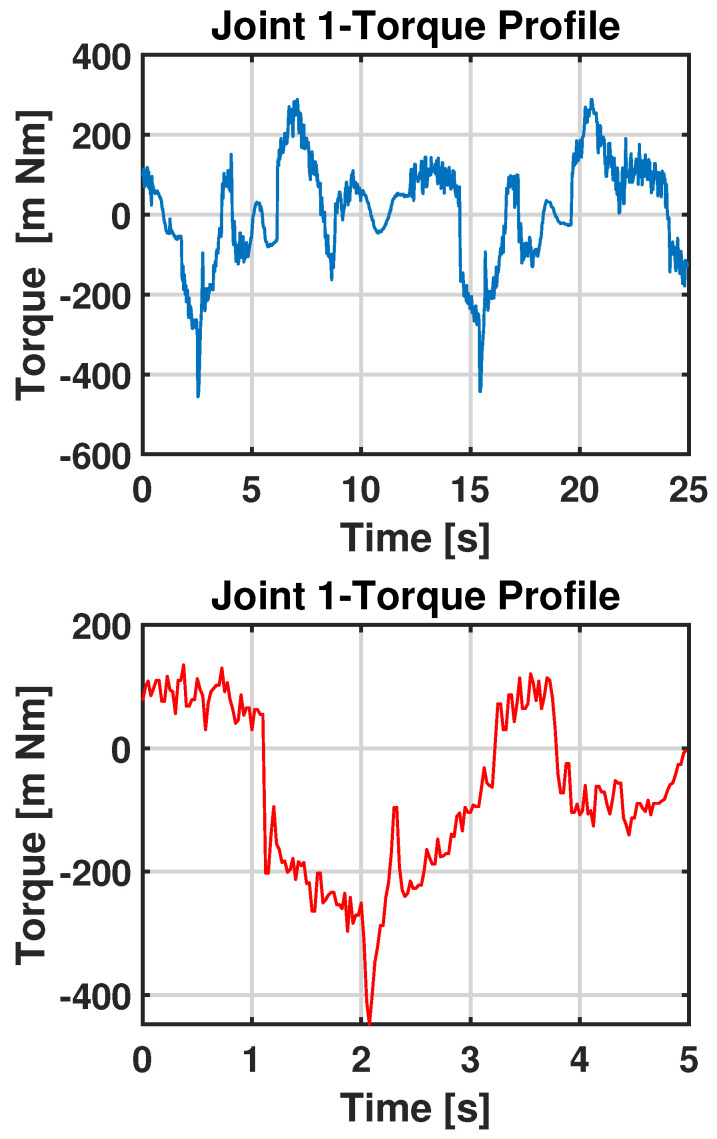
Torque profile of Joint 1: the blue plot represents the nominal data and the red plot represents the observation data.

**Figure 10 sensors-23-03260-f010:**
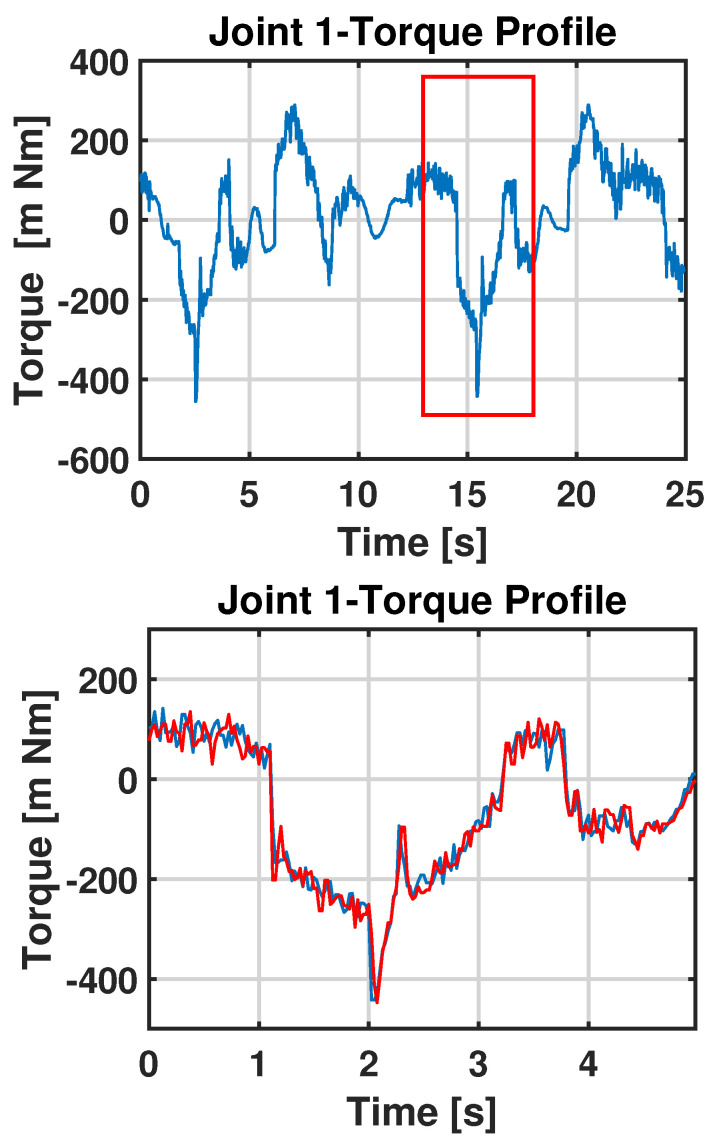
Comparison between the nominal data and the observation after synchronization.

**Figure 11 sensors-23-03260-f011:**
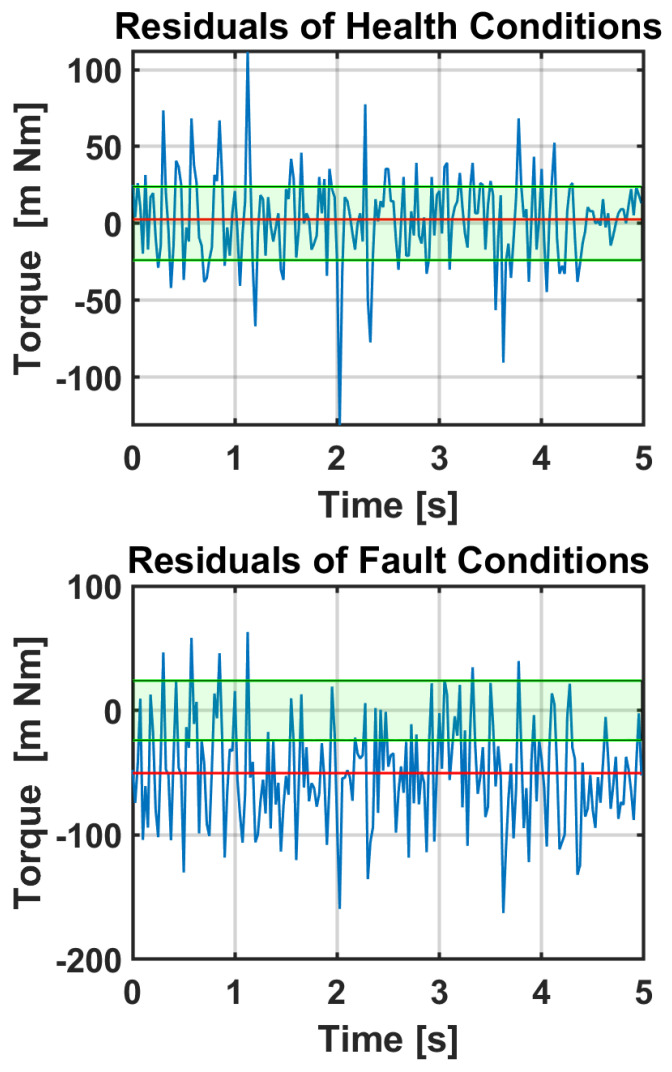
Residual plot under health and fault conditions. The red line represents the mean and the green band represents the healthy zone.

**Table 1 sensors-23-03260-t001:** Specifications of the collaborative manipulator tm5-900.

Field	Data
Reach (mm)	900
Maximum payload (kg)	4
Typical speed (m/s)	1.4
Angle of the joint 1, 6 (degree)	+/−270
Angle of the joint 2, 4, 5 (degree)	+/−180
Angle of the joint 3 (degree)	+/−155

## Data Availability

Not applicable.

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
