# Peer review of "Anomaly Detection and Concept Drift Adaptation for Dynamic Systems: A General Method with Practical Implementation Using an Industrial Collaborative Robot"

_sensors, 2023, doi:10.3390/s23063260_

Round 1

Reviewer 1 Report

The authors presented a relevant research in the field of cobots. Overall, the manuscript is well-structured and written. However, some portions of the text, listed below, require some improvement.  

Line 47-48. The sentence "Thus, algorithms for..." should be placed in a new paragraph, without bullets, and be connected with the upcoming paragraph (line 49 to 53).

Line 58. What is the meaning of CM?

Line 68. I would recommend the verb 'given' in place of "explained".

Line 69. I would recommend 'Section 2' in place of "Next Section".

In Figure 1, there is a typo in the first rectangle (anomaly detector).

Line 114. The acronym VAE was first defined in the manuscript in line 109.

Line 141. What is the meaning of ROS?

Line 155. Please revise the sentence starting with "In order to test...".

In Table 1, 'kg" in the right format for the unit of weight.

Line 175* Please revise the sentence starting with "Therefore has been added..."

Line 175* Please revise the sentence starting with "The first joint..."

Line 175 (entire paragraph). Please revise the writing of the whole paragraph. This very long paragraph in not clear and contains many english errors (some of them mentioned in the two items above, but there are much more).

Line 199. Please revise the sentence starting with "In Figure 6..."

Line 204 (and other instances). The unit of time is 's' (not "sec").

Reviewer 2 Report

This paper discusses the anomaly detection problem under concept drift (CD). As the authors mentioned, most existing work on anomaly detection is for bounded condition. Therefore, the work in this paper is of significant importance.

Generally, the paper is technically correct. The authors adopt a CD detector along with an Anomaly detector. Experimental results are presented based on a robot.

My suggestion is - for most anomaly detection work, some statistical results (i.e., Monte Carlo simulation) are also presented, since one experiment or realization is typically insufficient. The paper could be better if the authors can make it statistically sound. However, it is understandable that it is financially impossible to repeat the experiment many times. Therefore, this is optional.

Overall, the paper is well written. Publication is recommended upon minor improvements in English languages.

Reviewer 3 Report

General Comments:

1. The descriptions of the flows in Figure 1 are to be improved. For example, What are the differences between the two flows from the training stage to (1) the CD detection stage (labeled with CNN's weights) and to (2) the anomaly detection stage (labeled with Data acquisitions)?

2. How are the developed CNN detector models tested? How are the Proof-of-Concept experiments done? Please provide more descriptions on the validation and verification of the proposed models. 

Questions/Suggestions:

Line 3

Since Fault diagnosis ...

=> Since fault diagnosis ...

Line 4

where the working condition are ..

=> where the working conditions are ...

Lines 28-29

through the Anomaly detection (AD).

=> through anomaly detection (AD).

Line 41

or defects. Where the most ...

=> or defects where the most ...

Line 48

between a failure and a CD.

=> between a failure vs. a CD.

Line 76

Model-based [8][9] approach relies on ...

=> The model-based approach [8][9] relies on ...
